# Implicit Rank-Minimizing Autoencoder

**Li Jing**
Facebook AI Research
New York
ljng@fb.com

**Jure Zbontar**
Facebook AI Research
New York
jzb@fb.com

**Yann LeCun**
Facebook AI Research
New York
yann@fb.com

## Abstract

An important component of autoencoders is the method by which the information capacity of the latent representation is minimized or limited. In this work, the rank of the covariance matrix of the codes is implicitly minimized by relying on the fact that gradient descent learning in multi-layer linear networks leads to minimum-rank solutions. By inserting a number of extra linear layers between the encoder and the decoder, the system spontaneously learns representations with a low effective dimension. The model, dubbed Implicit Rank-Minimizing Autoencoder (IRMAE), is simple, deterministic, and learns compact latent spaces. We demonstrate the validity of the method on several image generation and representation learning tasks.

## 1 Introduction

Optimizing a *linear* multi-layer neural network through gradient descent leads to a low-rank solution. This phenomenon is known as implicit regularization and has been extensively studied under the context of matrix factorization [9, 1, 21], linear regression [24, 6], logistic regression [25], and linear convolutional neural networks [8]. The main goal of these prior works were to understand the generalization ability of deep neural networks. By contrast, the goal of the present work is to design an architecture that takes advantage of this phenomenon to improve the quality of learned representations.

Learning good representations remains a core issue in AI [2]. Representations learned in a self-supervised (or unsupervised) manner can be used for downstream tasks such as generation and classification. Autoencoders (AE) are a popular class of method for learning representations without requiring labeled data. The internal representation of an AE must have a limited information capacity to prevent the AE from learning a trivial identity function. Variants of AEs differ by how they perform this limitation. Bottleneck AE (sometimes called "Diabolo networks") simply use low-dimensional codes [23], noisy AE, such as variational AE add noise to the codes while limiting the variance of their distribution [4, 14], quantizing AE (such as VQ-VAE) quantize the codes into discrete clusters [27], sparse AE impose a sparsity penalty on the code [19, 20], contracting and saturating AE minimize the curvature of the network function in directions outside the data manifold [22, 7], and denoising AE are trained to produce large reconstruction error for corrupted samples [28].

In this work, we propose a new method to implicitly minimize the rank/dimensionality of the latent code of an autoencoder. We call this model Implicit Rank-Minimizing Autoencoder (IRMAE). This method consists in inserting extra linear layers between the encoder and the decoder of a standard autoencoder. This additional linear network is trained jointly with the rest of the autoencoder through classical backpropagation. As a result, the system spontaneously learns representations with a low effective dimensionality. Like other regularization methods, this extra linear neural network does not appear at inference time as the linear matrices collapse into one. Thus, the encoder and the decoder architecture of the model is identical to the original model. In practice, we fold the collapsed linear matrices into the last layer of the encoder at inference time.

We empirically demonstrate IRMAE's regularization behavior through a synthetic dataset and show that it learns good representation with a much smaller latent dimension. Then we demonstrate superior representation learning performance of our method against a standard deterministic autoencoder and comparable performance to a variational autoencoder on MNIST dataset and CelebA dataset through a variety of generative tasks, including interpolation, sample generation from noise, PCA interpolation in low dimension, and a downstream classification task. We also conducted an ablation study to verify that the advantage of implicit regularization comes from gradient descent learning dynamics.

We summarize our contributions as follows:

- We proposed a method of inserting extra linear layers in deep neural networks for rank regularization;
- We proposed a simple, deterministic rank-minimization autoencoder that learns low-dimensional representation;
- We demonstrated a superior performance of our method compared to a standard deterministic autoencoder and a variational autoencoder on a variety of generative and downstream classification tasks.

## 2   Related Work

The implicit regularization provided by gradient descent optimization is widely believed to be one of the keys to deep neural networks' generalization ability. Many works focusing on linear cases are trying to study this behavior empirically and theoretically. Soudry et al. [25] show that implicit bias helps to learn logistic regression. Saxe et al. [24] study a 2-layer linear regression and theoretically demonstrated that continuous gradient descent could lead to a low-rank solution. Gidel et al. [6] extend such theory to a discrete case for linear regression problems. In the field of matrix factorization, Gunasekar et al. [9] theoretically prove that gradient descent can derive minimal nuclear norm solution. Arora et al. [1] extend this concept to the deep linear network case by theoretically and empirically demonstrating that a deep linear network can derive low-rank solutions. Gunasekar et al. [8] prove that gradient descent has a regularization effect in linear convolutional networks. All these works are trying to understand why gradient descent can help generalization in existing approaches. On the contrary, we take advantage of this phenomenon to develop better algorithms. Also, the current implicit regularization study requires a small gradient and vanishing initialization, while our method is more general and can be used with complicated optimizers such as Adam [13] and allow combination with more complicated components.

Autoencoders are popular for representation learning. It is important to limit the latent capacity as the data are embedded in a lower-dimensional space. A big family of them are based on variational autoencoders [14] such as beta-VAE [12]. These methods tend to generate blurry images due to its intrinsic probabilistic nature. On the other hand, a naive deterministic autoencoder is considered a failure in generative tasks and has "holes" in its latent space, due to the absence of explicit constraint on the latent distribution. Many methods with deterministic autoencoder are proposed to solve this problem, such as RAE [5], WAE [26], VQ-VAE [27].

## 3   Implicit Rank-Minimizing Autoencoder

We denote by $\mathcal{E}()$ and $\mathcal{D}()$ the encoder and decoder of a deterministic autoencoder, respectively. The latent variable $z \in \mathbb{R}^d$ is determined by $\mathcal{E}(y)$. Encoder and decoder are classically trained by jointly minimizing the $L_2$ reconstruction loss $L_{AE} = ||y - \mathcal{D}(\mathcal{E}(y))||_2^2$. Without any constraint on the latent space, a simple deterministic autoencoder will typically learn a non-Gaussian latent space with "holes" and hence does not generate good samples.

Implicit rank-minimizing autoencoder consists in adding extra linear matrices $W_1, W_2, \cdots, W_l$ between the encoder and decoder, where $W_i \in \mathbb{R}^{d \times d}$ are randomly initialized. The corresponding diagram is shown in Figure 1. All $W_i$ matrices are trained jointly with the encoder and the decoder. Hence, the reconstruction loss is represented as

$$L = ||y - \mathcal{D}(W_l \cdots W_2 W_1 \mathcal{E}(y))||_2^2 \tag{1}$$

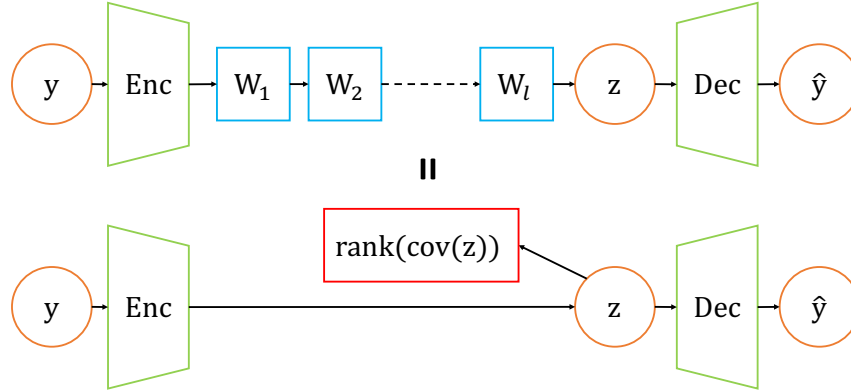

Figure 1: Implicit rank-minimizing autoencoder: a deterministic autoencoder with implicit regularization. The linear matrices that form a linear neural network between the encoder and the decoder are all square matrices. The effect of these matrices is to penalize the rank of the code variable. These matrices are equivalent to a single linear layer at inference time, and thus they do not change the capacity of the autoencoder. In practice, they are absorbed into the last layer of the encoder.

During training, these matrices encourage latent variables to use a lower number of dimensions and effectively minimize the rank of the covariance matrix of the latent space. Thus, one can amplify the regularization effect by adding more $W_i$ matrices between the encoder and the decoder. Also, we do not use special initialization of each $W_i$, and it works with more optimizers such as Adam [13].

During inference, all $W_i$ matrices can be "absorbed" into the encoder as all the linear matrices collapse, as linear matrix multiplication is associative. Therefore, we can directly use this linearly modified decoder for generative tasks; we can also directly use the encoder for downstream tasks such as classification.

## 4 Experiment

In this section, we empirically evaluate the proposed IRMAE model. We first verify the regularization effect through a synthetic task. We then demonstrate that IRMAE generates higher quality images compared to a baseline AE. IRMAE shows comparable performance to VAE. Lastly, we demonstrate IRMAE's superior performance on downstream classification tasks.

Throughout all the experiments, we demonstrate the latent dimension by plotting the normalized singular values. Each plot in Figures 2 and 4 depicts singular values (sorted from large to small) of the covariance matrix of the latent variables $z$ corresponding to examples in the validation set. The plots are normalized by dividing each singular value by the largest singular value of the covariance matrix. Therefore, the dimension of latent space can be interpreted as the number of nonzero singular values.

### 4.1 Verification with Known Intrinsic Dimension

We verify the regularization behavior of IRMAE via a synthetic shape dataset. Each example is a 32x32 RGB image with a random-color, random-sized square or circle, located at a random position. Hence, the data has a known intrinsic dimensionality of 7 (3 for color, 2 for coordinate, 1 for size, 1 for shape).

The base architecture we used is a deterministic autoencoder. The architecture and experimental detail can be found in supplementary material. We use a latent dimension of 32. For IRMAE, we use $l = 2$ and $l = 4$ extra matrices between the encoder and the decoder. We test our method against non-regularization, L1 regularization, and L2 regularization on the hidden code with the

same architecture. We demonstrate the learned latent space in Figure 2. The baseline model, L1 regularization, L2 regularization, IRMAE with $l = 2$ yields excellent reconstructions on validation set.

This result shows that IRMAE with $l = 2$ is able to learn good latent representation with a rank close to intrinsic dimension, while L1, L2 regularization tends to use a much larger latent space.

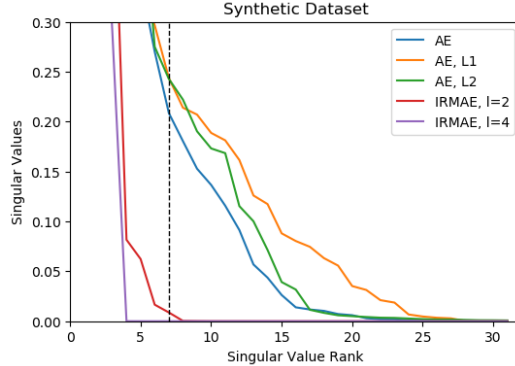

Figure 2: Singular values of the latent space of each model on synthetic shape dataset. Each curve represents singular values of the covariance matrix of the code computed on the validation set. IRMAE $l = 2$ is able to approach the minimal theoretical rank of 7.

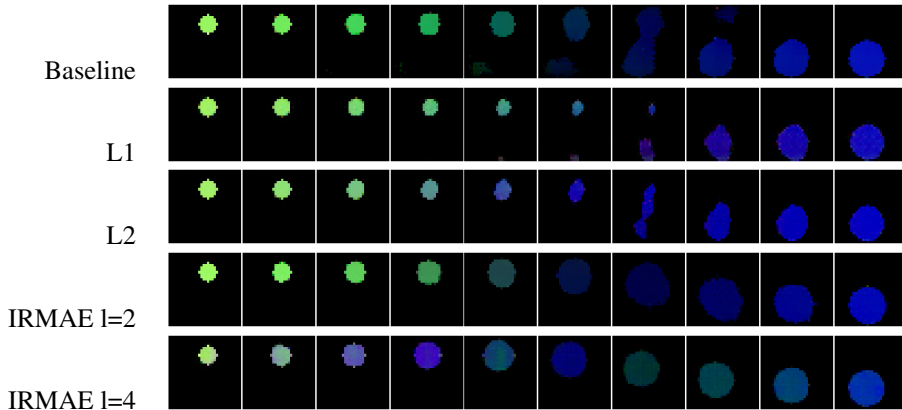

Figure 3: Linear interpolation between two randomly generated samples. From top to bottom are results from baseline unregularized AE, AE with L1 regularization, AE with L2 regularization, IRMAE $l = 2$, IRMAE $l = 4$.

### 4.2 Image Generation

Generating high-quality images by sampling the latent space is one of the key indicators of a good representation. In order to provide a comparison with standard deterministic autoencoders and variational autoencoders [14], we train our model on the MNIST dataset [15] and the CelebA dataset [16]. We set the latent dimension to 128/512 for the two datasets, respectively. We use 8/4 extra linear matrices for regularization in IRMAE, respectively. More experiment detail can be found in the supplementary material. We evaluate our model on a variety of representation learning tasks: interpolation between data points, sample generation from random noise, downstream classification task, PCA interpolation in latent space. We also quantitatively evaluate the sample generation by using the FID score. Each model uses the same architecture, except that the VAE code is twice as large to include the means and variances. On all these tasks, our method demonstrates comparable performances to the VAE.

**Latent Dimension** We show the latent dimensionality reduction of our method in Figure 4. IRMAE utilizes significantly lower-dimensional latent space compared to baseline autoencoder. Notice that

we omit the VAE's curve because VAE uses the whole latent space and hence all singular values tend to be large.

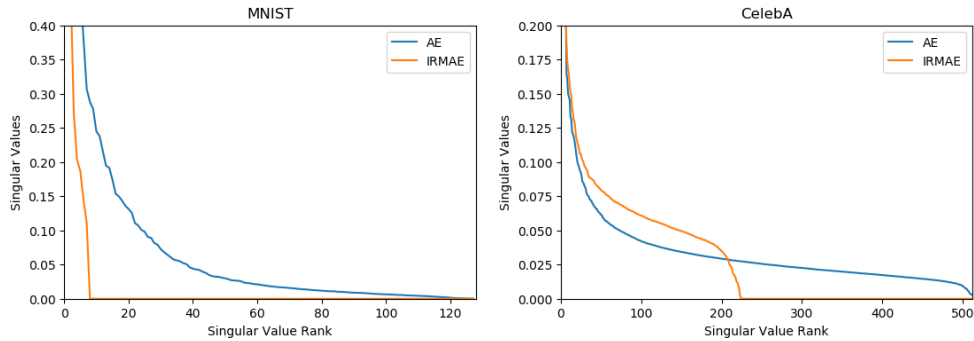

Figure 4: Singular value spectra of covariance matrices of codes for MNIST and CelebA datasets by IRMAE and a baseline AE. Each curve represents the singular values of the covariance matrix of the hidden code computed on the validation set.

**Interpolation between Data Points:** We linearly interpolate the latent variable between two images from the validation set. The generated results are shown in Figure 5. IRMAE significantly outperforms the baseline AE on MNIST.

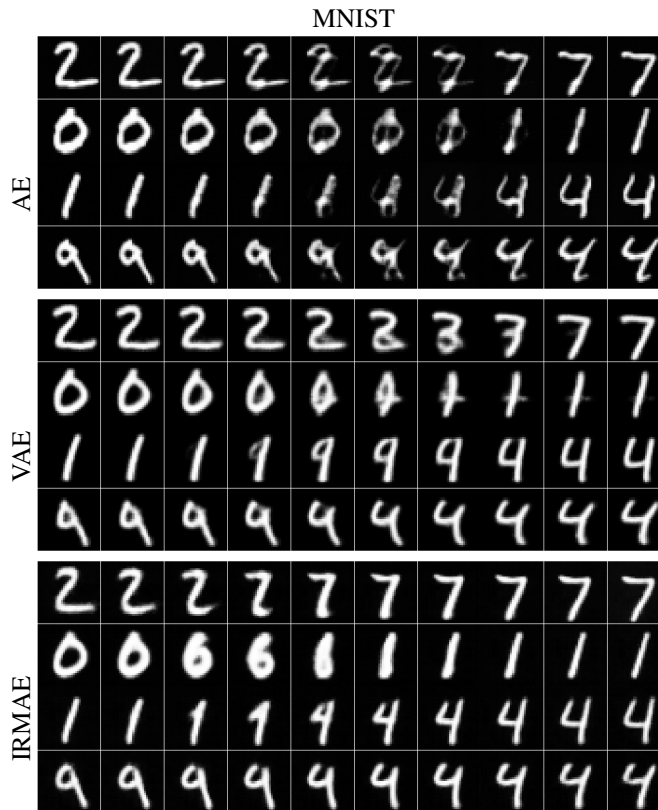

Figure 5: Linear interpolation between data points on the MNIST dataset. From top to bottom are images generated from an unregularized AE, a VAE, and an IRMAE, respectively. IRMAE produces higher quality images.

**Sampling from Noise:** Deterministic autoencoders are not considered to be generative models. It is essential to have constraints on the latent space to derive such ability [2]. Here, we show that IRMAE can sample high-quality images from Gaussian noise. Specifically, we sample random latent variables

from 1) a multivariate Gaussian captured by this covariance matrix, 2) a Gaussian Mixture Model with 4/10 clusters. The generated results are presented in Figure 6 and Figure 7. We quantitatively evaluate the performance of each model by using the Frechet Inception Distance (FID) [11] and report the results on MNIST/CelebA in Table 1.

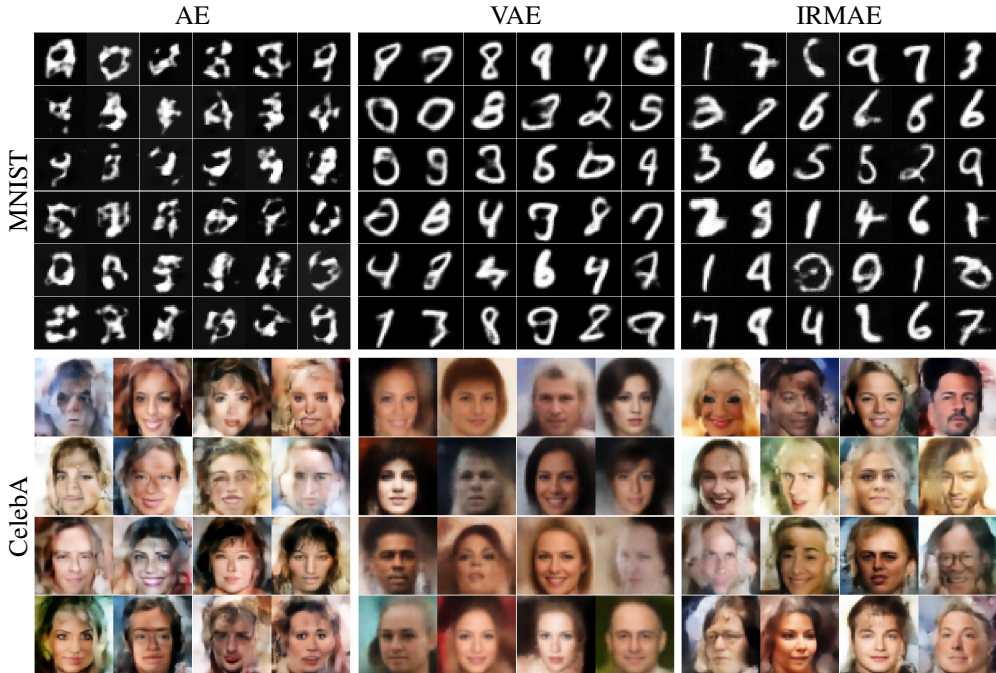

Figure 6: MNIST/CelebA images samples from Multivariate Gaussian with covariance estimated from training set. From left to right are images generated from an unregularized AE, a VAE, and an IRMAE, respectively.

Table 1: FID score (smaller is better) for samples of various models for MNIST/CelebA.

| | Multivariate Gaussian | | | | Gaussian Mixture Model | | |
|---|---|---|---|---|---|---|---|
| | AE | VAE | IRMAE | | AE | VAE | IRMAE |
| MNIST | 55.0 | 33.9 | 37.4 | MNIST | 38.0 | 30.8 | 34.0 |
| CelebA | 52.8 | 51.8 | 42.5 | CelebA | 49.0 | 48.8 | 36.4 |

**PCA on Latent Space:** We verify that IRMAE learns a compact and continuous latent space by performing PCA on the latent space. We project all latent variables to a 2-dimensional space. We randomly sample vectors in this low dimensional space and interpolate them along two principal vectors. The corresponding images are sampled from inverse PCA followed by the decoder, which is shown in Figure 8. IRMAE generates higher quality images compared to VAE.

Additional experiments are demonstrated in the supplementary material, including comparing IRMAE to other deterministic AEs, comparing IRMAE against AEs with various latent dimension, effect of varying linear layer depth in IRMAE.

## 4.3 Downstream classification

Latent variables are useful for downstream tasks since they capture the main underlying structure of the data distribution [10, 18, 3]. These self-supervised learning methods have the exciting potential to outperform purely-supervised models. We train a multilayer perceptron head on the latent variable generated by the encoder, to classify MNIST images. This MLP head has two linear layers of hidden dimension 128, with ReLU activation. Thus, all models share the same architecture. Each model is trained with an Adam optimizer with a learning rate of 0.001. Early stopping is performed based on

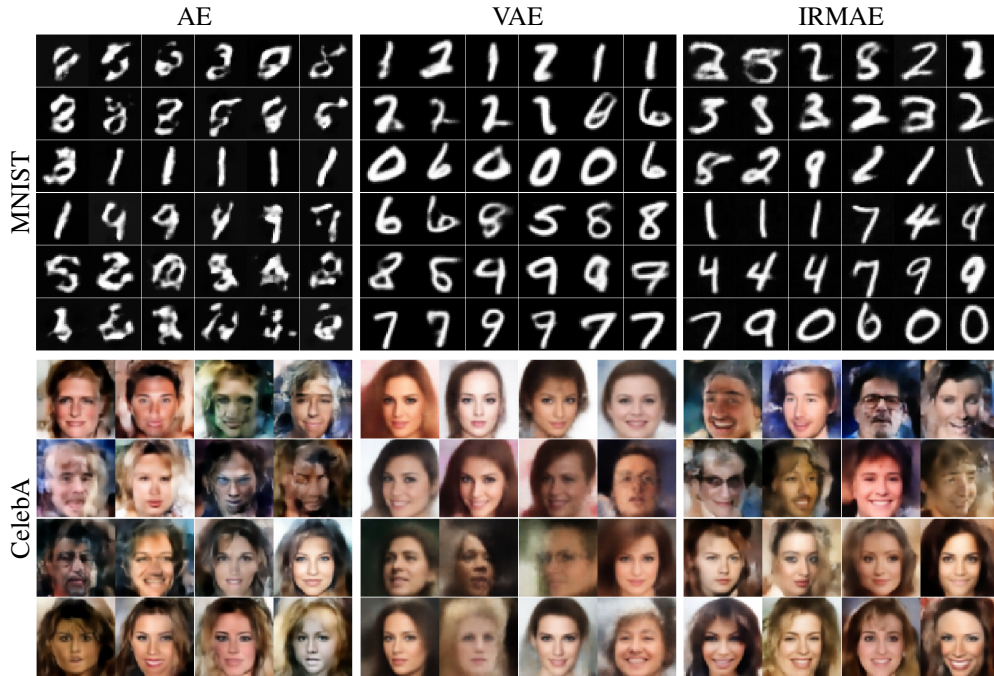

Figure 7: MNIST/CelebA images samples from Gaussian Mixture Model with 4/10 clusters. From left to right are images generated from an unregularized AE, a VAE, and an IRMAE, respectively.

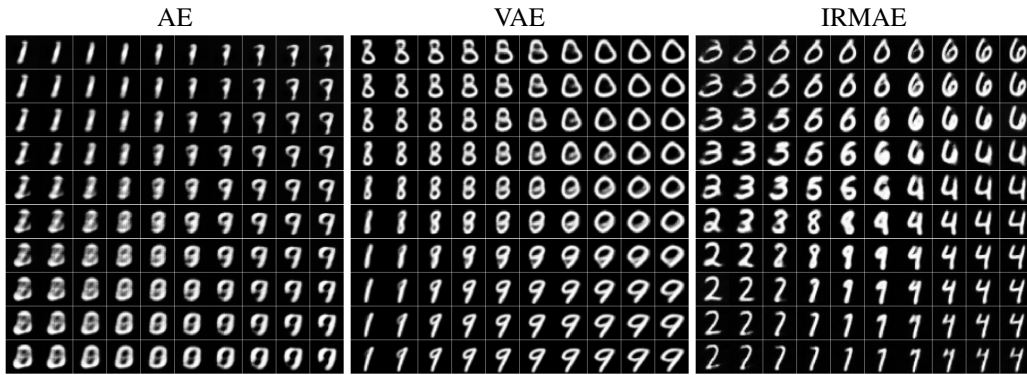

Figure 8: Sampling images from 2-dimensional space, mapped by PCA from latent variables. We interpolate along two principal components to generate samples. From left to right are images generated from an unregularized AE, a VAE, and an IRMAE, respectively.

validation set accuracy. The encoder weights are kept fixed. We compare our method against several baselines as well as the supervised version whose entire network is trained jointly. Representations learned by IRMAE obtain a significantly lower error rate compared to those from the unregularized AE in this task. The results are listed in Table 2.

## 4.4 Ablation Study

We perform several ablation studies to verify that the effect of dimensionality reduction comes from the extra linear neural network and its optimization dynamics.

**Linear matrices fixed:** In this ablation experiment, we fix the linear matrices to verify that the regularization effect comes from the learning dynamics instead of just the architecture. Figure 9 shows that under this condition, the regularization effect is weakened, and the sampled images are significantly worse.

Table 2: Downstream classification on MNIST dataset. We add a MLP head on top of the pretrained encoder by each method. Thus, all models share the same architecture. We do not perform fine tuning on the pretrained encoder except with the purely supervised version. Representation learned by IRMAE obtains significantly lower error rate compared to baselines and supervised version in the low labeled data regime.

| total training size | 10 | 100 | 1000 | 10000 | 60000 |
|---|---|---|---|---|---|
| AE | 31.4±0.5 | 30.2±0.3 | 10.6±0.2 | 3.7±0.1 | 1.9±0.1 |
| VAE | 21.8±1.0 | 21.7±0.4 | 5.1±0.2 | **1.7±0.1** | 1.1±0.1 |
| IRMAE | **12.0±0.9** | **10.2±0.5** | **3.8±0.3** | 2.4±0.2 | 1.9±0.1 |
| supervised | 29.1±2.6 | 25.1±0.6 | 6.0±0.4 | **1.7±0.1** | **0.8±0.1** |

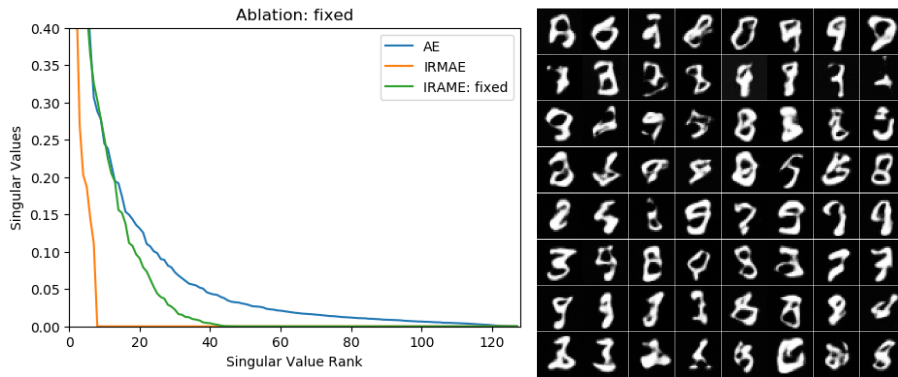

Figure 9: Ablation study: linear matrices fixed. This proves that the regularization behavior is not an effect of naive soft bottleneck.

**Nonlinearity between matrices:** One may suspect that the regularization effect comes from a deeper architecture. If we add nonlinearity between matrices, the model is equivalent to a standard autoencoder, with more layers. We show that adding a nonlinearity results in worse generation results, and the regularization effect is also completely lost. See Figure 10.

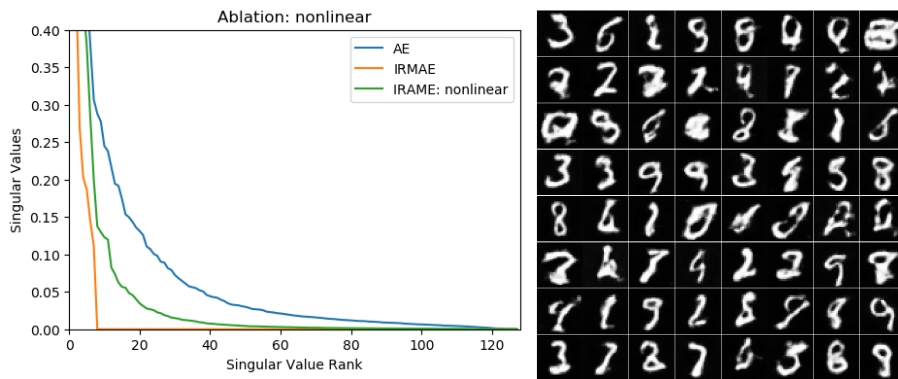

Figure 10: Ablation study: adding nonlinearity layers between linear matrices. This proves that the regularization behavior is not a naive effect of deeper architecture.

**Weight Sharing:** As our method introduces more parameters for training, it would be desirable to have all inserted matrices to share weights to reduce memory requirement. We show that forcing all matrices to share weights results in slightly worse generation results and weakened regularization effect. See Figure 11.

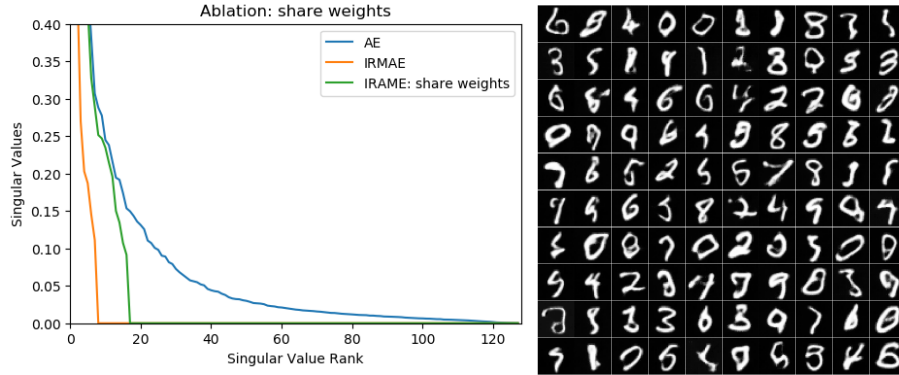

Figure 11: Ablation study: sharing weights in the inserted linear layers.

# 5 Conclusion

An important component of autoencoder methods is the method by which the information capacity of the latent representation is minimized or limited. In this work, the rank of the covariance matrix of the codes is implicitly minimized by relying on the fact that gradient descent learning in multi-layer linear networks leads to minimum-rank solutions. By inserting a number of extra linear layers between the encoder and the decoder, the system spontaneously learns representations with a low effective dimension. The model, dubbed Implicit Rank-Minimizing Autoencoder (IRMAE), is simple, deterministic, and low-rank latent space. We demonstrate the validity of the method on several image generation and representation learning tasks.

## Broader Impact

This work provides a novel approach to representation learning and self-supervised learning. It has the potential of boosting general self-supervised learning performances with social benefits including requiring less human data labeling, reducing power consumption of AI models, improving data privacy.

## Acknowledgement

We are grateful to Stephane Deny for his feedback on early versions of the manuscript. We thank Pascal Vincent, Nicolas Ballas, Lluis Castrejon, Piotr Bojanowski for their fruitful discussions.

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
