[Supplementary Material]

# Implicit Rank-Minimizing Autoencoder
## Supplementary Materials

## 1  Experiment Detail

### 1.1  Dataset

For the synthetic shape dataset, we generate shape images on the fly. The size of each shape is uniformly sampled between 3 and 8, inclusively. The color is uniformly sampled in RGB. The coordinate of the center of the shape is randomly sampled with x and y between 8 and 24, inclusively.

For the MNIST dataset, all images are resized to 32x32.

For the CelebA dataset, all images are center-chopped to 148x148 and then resized to 64x64.

### 1.2  Architecture

The architecture of the encoder and the decoder for each experiment is listed in Tables 1. $\text{Conv}_n$/ConvT denotes a convolutional/transposed-convolutional layer with the output channel dimension equal to $n$. All convolutional layers use 4x4 kernel size with a stride 2, padding 1. $\text{FC}_n$ denotes a fully connected network with output dimension $n$.

Table 1: The architecture of the encoder and the decoder for each experiment.

| Dataset | Shape | MNIST | CelebA |
|---|---|---|---|
| Encoder | $x \in \mathcal{R}^{32x32x3}$ <br> $\to \text{Conv}_{32} \to \text{ReLU}$ <br> $\to \text{Conv}_{64} \to \text{ReLU}$ <br> $\to \text{Conv}_{128} \to \text{ReLU}$ <br> $\to \text{Conv}_{256} \to \text{ReLU}$ <br> $\to \text{Conv}_{32} \to \text{ReLU}$ <br> $\to z \in \mathcal{R}^{32}$ | $x \in \mathcal{R}^{32x32x1}$ <br> $\to \text{Conv}_{32} \to \text{ReLU}$ <br> $\to \text{Conv}_{64} \to \text{ReLU}$ <br> $\to \text{Conv}_{128} \to \text{ReLU}$ <br> $\to \text{Conv}_{256} \to \text{ReLU}$ <br> $\to$ flattern_to 1024 <br> $\to \text{FC}_{128} \to z \in \mathcal{R}^{128}$ | $x \in \mathcal{R}^{64x64x3}$ <br> $\to \text{Conv}_{128} \to \text{ReLU}$ <br> $\to \text{Conv}_{256} \to \text{ReLU}$ <br> $\to \text{Conv}_{512} \to \text{ReLU}$ <br> $\to \text{Conv}_{1024} \to \text{ReLU}$ <br> $\to$ flattern_to 16384 <br> $\to \text{FC}_{512} \to z \in \mathcal{R}^{512}$ |
| Decoder | $z \in \mathcal{R}^{32}$ <br> $\to \text{ConvT}_{256} \to \text{ReLU}$ <br> $\to \text{ConvT}_{128} \to \text{ReLU}$ <br> $\to \text{ConvT}_{64} \to \text{ReLU}$ <br> $\to \text{ConvT}_{32} \to \text{ReLU}$ <br> $\to \text{ConvT}_{3} \to \text{Tanh}$ <br> $\to \hat{x} \in \mathcal{R}^{32x32x3}$ | $z \in \mathcal{R}^{128}$ <br> $\to \text{FC}_{8096}$ <br> $\to$ reshape_to 8x8x128 <br> $\to \text{ConvT}_{64} \to \text{ReLU}$ <br> $\to \text{ConvT}_{32} \to \text{ReLU}$ <br> $\to \text{ConvT}_{3} \to \text{Tanh}$ <br> $\to \hat{x} \in \mathcal{R}^{32x32x1}$ | $z \in \mathcal{R}^{512}$ <br> $\to \text{FC}_{65536}$ <br> $\to$ reshape_to 8x8x1024 <br> $\to \text{ConvT}_{512} \to \text{ReLU}$ <br> $\to \text{ConvT}_{256} \to \text{ReLU}$ <br> $\to \text{ConvT}_{128} \to \text{ReLU}$ <br> $\to \text{ConvT}_{3} \to \text{Tanh}$ <br> $\to \hat{x} \in \mathcal{R}^{64x64x3}$ |

For VAE models, the last layer of the decoder has doubled output dimension, which is split as the average and the standard deviation. It also uses Sigmoid instead of Tanh.

## 1.3 Hyperparameters

The following hyperparameters for each experiment are listed in Table. 2. The number of epochs is chosen for converged reconstruction error for the base model.

Table 2: hyperparameters.

| Dataset | Shape | MNIST | CelebA |
|---|---|---|---|
| learning rate | 0.0001 | 0.0001 | 0.0001 |
| epochs | 100 | 50 | 100 |
| latent dimension | 32 | 128 | 512 |
| batch size | 32 | 32 | 32 |
| training examples | 50000 | 60000 | 162770 |
| evaluation examples | 10000 | 10000 | 19962 |

# 2 Additional Experiments

## 2.1 Effect of Varying Linear Layers Initial Variance

Initial variance of the linear matrices has strong influence on the regularization effect. We observe that a larger variance weakens the regularization effect. See Table.3.

Table 3: Effect of varying initial variance of linear layers in IRMAE. Performed on MNIST dataset. Latent rank represents corresponding number of nonzero singular values of the covariance matrix of latent space.

| Variance | 1x | 2x | 4x |
|---|---|---|---|
| Latent Rank | 8 | 43 | 66 |
| FID | 37.4 | 33.8 | 49.0 |

## 2.2 Effect of Varying Linear Layers Depth

Adding more linear layers will increase the regularization effect. We demonstrate such effect in Table.4. The number of linear layers $l$ is a hyperparameter and needs to be optimized in practice.

Table 4: Effect of varying linear layers depth. Performed on MNIST dataset. Latent rank represents corresponding number of nonzero singular values of the covariance matrix of latent space.

| Depth (l) | 2 | 4 | 8 | 12 |
|---|---|---|---|---|
| Latent Rank | 70 | 39 | 8 | 4 |
| FID | 44.0 | 30.1 | 37.4 | 62.6 |

## 2.3 Comparing to State-of-the-art Deterministic AEs

We compare IRMAE against several modern deterministic autoencders including WAE and RAE. IRMAE demonstrates superior performance on CelebA dataset. See Table.5.

Table 5: Comparing IRMAE against state-of-the-art deterministic AEs on CelebA dataset.

| WAE [3] | RAE [1] | IRMAE |
|---|---|---|
| 53.7 | 44.7 | **42.0** |

## 2.4 Comparing to AEs with Various Latent Dimension

Autoencoders with different latent dimension or prior setting has trade-off in learning useful representations. Here, we study the effect of latent dimensionality of IRMAE against AE in Table.6 and Figure.1. IRMAE with larger latent dimensions outperforms the optimal dimensional AE.

Table 6: Comparing IRMAE against AEs with different latent dimension. Performed on CelebA dataset. IRMAE uses $l = 4$ throughout the experiment. Results are listed in FID score.

| Latent dimension | 32 | 64 | 128 | 256 | 512 |
|---|---|---|---|---|---|
| IRMAE ($l = 4$) | 81.6 | 64.6 | 47.6 | 42.7 | 42.0 |
| AE | 78.2 | 60.1 | 46.0 | 45.4 | 53.9 |

Figure 1: IRMAE vs AE with varying latent dimension

Figure 1: Comparing IRMAE against AEs with different latent dimension. Performed on CelebA dataset.

## 3 t-SNE visualization

We visualize the density of the sampled MNIST images by each model in Figure 2 using t-SNE [2]. Blue points represent the original data point, and the orange points represent the sampled ones. We compare IRMAE against an AE and a VAE. It's desirable that two point-clouds overlap. IRMAE demonstrates a comparable performance to VAE and a superior performance to AE.

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

multivariate Gaussian                    GMM

Figure 2: t-SNE visualization on MNIST images. Blue points represent the test set data point. Orange points represent the sampled images.