[Reviews · NeurIPS 2020]

Review 1

Summary and Contributions: This paper presents a new autoencoder method, the implicit rank-minimizing autoencoder. To enforce an approximate low rank constraint on learned latent codes, the method adds several linear layers in between a nonlinear encoder and decoder. The dynamics of gradient descent naturally learn low rank solutions in this setting, as demonstrated theoretically in prior work, and verified here through synthetic experiments. After learning, the added layers can be collapsed back to a single matrix. The resulting autoencoder method is evaluated on several image generation tasks, where it performs favorably/comparably to a VAE. The method also provides a strong boost to semisupervised learning, particularly in the low data regime. The contributions are: - A simple regularization method to enforce a low rank constraint on autoencoder codes (or in deep networks and other achitectures more generally) - Experimental results showing that this method yields performance improvements in a range of contexts relative to unregularized autoencoders and variational autoencoders

Strengths: This paper is quite amazing. Just adding a few linear layers causes otherwise standard, deterministic autoencoders to learn interesting generative factors of a similar or possibly greater quality to VAEs. It's rare to see a simple idea that works very well, with many possible extensions. I think this result will be of wide interest to the community. The theoretical observation that gradient descent dynamics in deep linear networks finds low rank solutions is well established, but has not been put to practical use. This paper is highly original in finding an impactful way of doing so. The empirical evaluation shows that the proposed method is comparable and possibly superior in certain cases to VAEs, and much superior to unregularized deterministic autoencoders. First let me caveat that I am not an expert in current best practices for evaluating generative models of this sort. My reading of the evaluation is that it is reasonably thorough, and shows that the proposed method is at least on par with VAEs. To me, this is enough for this paper to be clearly above bar for NeurIPS--it takes a very different route to achieve similar results to a widely used method, and is likely to spur a variety of follow up work which will improve performance in the future. The most decisive advantage for the method is in the semisupervised setting, where it achives a marked reduction in classification error at small training set sizes. This is a highly compelling result. The synthetic experiments verify that adding layers does impart a low rank bias, and make the nice point that a low rank constraint is not achievable with popular simple regularizers like L1/L2 regularization. These experiments provide good evidence that the technique works for the reason given. -----Update after Rebuttal------ Congratulations to the authors for a great paper. It proposes a method that all reviewers think is highly original and could lead to substantial follow ups. It is a genuinely different approach to building a generative model. And it is remarkably simple. The criticisms, particularly post rebuttal, focus on the extensiveness of the empirical evaluation, not its correctness. In my opinion, empirical results should be used to test hypotheses, and there is no specific set of benchmarks that a paper must address to be accepted. This paper does not claim that it works better than all proposed generative models, and it does not need to. It is already interesting (to me at least) that such a simple method is competitive with VAEs. Asking for a new method that starts from a very different mechanism to immediately match state of the art performance relative to all other approaches is not necessary in my opinion. Future work will build on this paper to improve performance. The contribution of the work is showing that a simple method, that lies in a very different direction from prior work, performs about as well.

Weaknesses: One important hyperparameter is not discussed, namely, the initialization variance of the added linear layers. These weight matrices will presumably need to be initialized with small enough random weights such that their product is near zero before training in order for the low rank bias to be present. If the network were instead initialized with large weights, there would no longer be an implicit rank constraint (the linear layers would operate in the lazy regime). I could not find the initialization scheme in the main text or supplement. Overall I think the text could be easily clarified to state that the low rank strategy relies on 1) added linear layers, 2) gradient descent dynamics, and 3) 'small' initialization (i.e. the rich learning regime.)

Correctness: The claims and methods appear to be correct, but I am not an expert in evaluations of generative models.

Clarity: The paper is overall clear and easy to follow. Some minor typos (differe ->differ, ln 23)

Relation to Prior Work: The analysis in [22] also studies deeper linear networks of arbitrary depth, and shows that these have the same low rank bias (so the l=2 case is sufficient, and adding more layers would be predicted to behave relatively similarly). The second paragraph of the related work section could be improved slightly to directly contrast this prior work with the proposed method to highlight the differences.

Reproducibility: Yes

Additional Feedback: The supplement investigates a weight tying scheme. It is not fully clear what was tried here, but one note is that, based on the prior theoretical work, tying weights to their transpose should impart a similar bias. Eg, W_1=W and W_2=W^T


Review 2

Summary and Contributions: Building on recent theoretical works suggesting that gradient descent (GD) over linear neural networks (LNNs) exhibits an implicit regularization towards low rank, the paper proposes to equip autoencoders (AEs) with a LNN between the encoder and the decoder. This LNN is trained jointly with the AE, and its implicit regularization effectively leads to a low dimensional, and thus semantically meaningful, latent representation. At test time, the LNN is equivalent to a single linear operation which may be absorbed into the encoder. Various experiments compare the proposed method --- named Implicit Rank-Minimizing Autoencoder (IRMAE) --- to standard AE and variational AE (VAE), demonstrating superior performance on image generation and representation learning tasks.

Strengths: The topic of implicit regularization in deep learning is extremely relevant to the NeurIPS community, and the idea of using a theoretical model used to study this phenomenon for practical gains is refreshing, and contributes to the discourse between practice and theory. I also found the paper clear and easy to read.

Weaknesses: Given that the contribution in this work is solely empirical, in my opinion, the experimental evaluation should be much more comprehensive. In particular, I believe the authors should either compare IRMAE to a wide array (including state of the art) AEs on various modern benchmarks. At present, IRMAE is only compared to a basic AE and VAE. Moreover, in many of the experiments, the AE has latent dimension equal to the input dimension, rendering it completely incapable. I believe it is necessary to evaluate the AE with varying latent dimension. While IRMAE does have the advantage of not requiring prior knowledge regarding the appropriate latent dimension, a-priori, it could be that a correct choice of the latter would render the AE superior to IRMAE. This point is merely an example which I use to illustrate my claim that the empirical evaluation in this work (its primary contribution) should be much more extensive. I would also like to note that many important details regarding the experiments that have been made are deferred to an appendix (for example the architecture of the AE), which in my opinion does not do justice given their centrality. I recommend including these details in the body of the paper, and if needed, deferring some of the (many, and oftentimes repetitive) figures to the appendixes. === UPDATE FOLLOWING AUTHOR FEEDBACK === I have thoroughly read the authors' feedback, as well as the other reviews. I agree with the other reviewers (R1 in particular) that the idea of harnessing the implicit regularization in linear neural networks for autoencoding is promising, and the authors make a strong case for its viability. At the same time, while the authors' response does begin to address my (and other reviewers') concerns regarding soundness of their empirical evaluation, my feeling is that the required modifications warrant an additional review cycle. I encourage the authors to revise the manuscript and submit to another machine learning venue, as I believe this work may be of great interest to the community. I have slightly increased my overall score to account for the above.

Correctness: The claims and methods are correct as far as I could tell, but the empirical methodology is lacking in my opinion (see above).

Clarity: Yes

Relation to Prior Work: Related work is generally referenced properly. I would recommend however discussing other empirical works that made use of linear neural networks for practical gains, for example "Blind Super-Resolution Kernel Estimation Using an Internal-GAN" by Bell-Kligler et al.

Reproducibility: Yes

Additional Feedback: Semantic comments: * Line 56: should be "E() and D()", not "D() and E()". * Line 79: "Through out" ==> "Throughout" * Line 92: It is not clear that "l=2 and 4" means "l=2 and l=4" * Line 97: "much a larger" ==> "a much larger" * Line 114: "Point" ==> "Points" * Line 121: "Figure" ==> "Figures" * Figure 9 caption: "a effect" ==> "an effect" * Line 170: "these method tends" ==> "these methods tend" * Line 173: "is proposed" ==> "are proposed"


Review 3

Summary and Contributions: The authors propose to improve the quality of representations learned by an autoencoder by penalizing the rank of the learned representations. They do this without using any explicit regularization but by learning an additional neural network with linear layers between its encoder and decoder and training the entire setup together using classical backpropagation. The authors claim that representations learned in this way perform better than a standard deterministic autoencoder and comparable to variational autoencoders on generative tasks like interpolation, sample generation from noise and downstream classification tasks. This work is based on the assumption and some previous studies of implicit regularization-that optimizing a linear multi-layer neural network through gradient descent leads to a low-rank solution. The algorithm is tested on a synthetic dataset, and on MNIST and CelebA datasets through generative tasks and a downstream classification task.

Strengths: This seems to be one of the first works to use implicit regularization of linear neural networks for learning better representations. Their empirical evaluation suggests that this architecture performs better at generative tasks like interpolation and generation than vanilla autoencoders.

Weaknesses: In this work, the authors have not provided justification for using their proposed network for minimum rank solutions than explicitly minimizing the rank using nonlinear autoencoders with a lower dimensional latent code. Previous works on implicit regularization have been done to better understand the generalizability of the solutions found via backpropagation of underdetermined systems. However, the advantages of using this model over the state of the art models for learning representations are not clear. The authors claim that the effect of regularization can be amplified by adding more linear layers between the encoder and the decoder. This claim is also supported by [1] - "depth leads to more accurate completion of a low-rank matrix when the number of observed entries is small" which in this case translates to better generalization with more linear layers. However, the author's experiments show that using 2 linear layers gives better reconstructions than using 4 linear layers. If the number of linear layers to be used is indeed a hyperparameter then it is unclear about how it should be chosen when the theoretical minimum rank is not known. The authors have in their rebuttal addressed some of these concerns but not all. As a result I increased my score.

Correctness: The claims made in this paper are supported by limited empirical evaluations. There are no theoretical claims. The empirical methodology is qualitative.

Clarity: The overall paper is reasonably well-written but with several obvious typos spread over the paper. Some of the details such the degree of shared weights amongst the linear layers are missing from the ablation studies. For example, in line 143 they mention that they fix the linear matrices. In this case it is unclear whether the number of linear matrices fixed or their weights are fixed. If their weights are fixed, are they fixed to the random initializations or are they learned in some way. The section on ablation studies is unclear and hence does not seem to be well motivated.

Relation to Prior Work: The ablation studies do not add any new information to the existing literature on implicit regularization. Also the author’s claims about their proposed architecture giving the desired results “with all types of optimizers “ seems theoretically and empirically unsupported.

Reproducibility: Yes

Additional Feedback: It is not clear how the proposed model fares against state of the art results of sparse autoencoders like contractive autoencoders or beta-VAEs, for generative tasks and downstream classifications tasks. The advantages of using this model over the others should be made more explicit. More details on how to choose the number of linear layers for different tasks should be provided. Questions like- how do the number of linear layers change when the task is image generation from noise versus downstream classification tasks should be addressed and answered. The section on using PCA in the latent space for interpolation is not well justified since for vanilla autoencoders or VAEs, the rank of the learned representations is higher and PCA will essentially capture only some of its variance. Whereas in the implicit regularization case, the interpolation is expected to give better results with PCA given that PCA would be able to capture most of the variance of a lower rank representation. The advantage of using this model instead of regular autoencoders to work as a generative model while giving good reconstructions is somewhat justified. However, it is unclear about the aspects in which this model performs better with respect to VAEs. I think these points should be addressed in the paper.


Review 4

Summary and Contributions: Building on recent work on implicit regularisation in deep matrix factorisation leading to low rank solutions, a variant to the (standard) Autoencoder (AE) with implicit minimisation of the rank of the latent code is proposed: the Implicit Rank-Minimising Autoencoder (IRMAE). IRMAE is qualitatively and quantitatively compared to AE and VAE on a synthetic shape dataset, MNIST and CelebA, displaying competitive results. The main contribution of the paper consists in introducing and describing the proposed approach.

Strengths: # Soundness: - The claims of providing a rank-minimising AE which leads to low dimensional latent space representations as compared to the standard AE (with additional regularisation) is empirically validated on a synthetic data set of objects of different shape, size, colour, and location, where the intrinsic dimensionality is known (7), as well as MNIST and CelebA. The corresponding results for the VAE, however, appear less compelling (see comments in “weaknesses” below) - With respect to sampling the latent space / generation and interpolation, comparable performance to VAEs on MNIST and CelebA are achieved, where in some cases IRMAE achieves qualitatively and quantitively (measured by Fréchet Inception Distance) better results than VAE (and AE) # Significance & Novelty, Relevance: - The proposed approach builds on recent insight on implicit regularisation and rank-minimisation in training deep linear networks and combines the idea with AEs to achieve low dimensional latent spaces effectively. As such, the model appears novel and potentially could be useful in related AE approaches to achieve low dimensional representations with yet another way of (implicit) regularisation.

Weaknesses: # Soundness: - With respect to statements: 1. Lines 76-77: “IRMAE shows comparable performance to VAE while using a smaller intrinsic dimension of the latent space.” 2. Lines 107-109: “On all these tasks, our method demonstrates comparable performances to the VAE while using a much lower effective-dimension code space.” In my opinion, the claim that IRMAE requires a smaller intrinsic dimension than VAE is not really demonstrated. As the authors write in lines 111-113, “[n]otice that we omit the VAE’s curve because VAE uses the whole latent space and hence all singular values tend to be large.” By model design and using a spherical prior, the VAE maps into the whole latent space and leads to singular values of similar size. Still, it might very well be that a setting with lower dimensional latent space in the VAE could perform equally well, as already the standard AE results suggest that up to 128/512 latent dimension might not be required for MNIST/CelebA. In my opinion, a fair comparison to justify the claim would require to gradually change the dimensionality of the latent space in the VAE setting to compare to intrinsic dimensions identified with IRMAE. Having both methods attain the same (or similar) reconstruction loss, one can study the minimal dimensionality of the latent space required for the VAE and compare it to the intrinsic dimensions obtained with IRMAE. - With respect to the statement in L. 131-132: “These self-supervised learning methods have the exciting potential to outperform purely-supervised models.” This claim seems a little bit optimistic, as the results in table 2 suggest the opposite for full MNIST. The supervised IRMAE performs best on the full dataset as well as on a training subset of size 10 000. For smaller training sizes, IRMAE seems to perform better. However, it is unclear how the subsets for training were chosen and what the stated relative errors really depict (average over several bootstraps?). Furthermore, the reduced training size is not motivated and it remains unclear why very small training sets of <= 1000 are relevant, in particular. Here, some more discussion of the results seems appropriate. - In addition, another relevant comparison in table 2 could be the supervised VAE setting (i.e. the Variational Information Bottleneck model by Alemi et al.) which might yield even better classification results. # Significance & Novelty, Relevance: - In general, the submission demonstrates convincingly that IRMAE is an interesting model and has beneficial properties, in particular compared to AEs. But some more discussion of the aspects raised with regards to the VAE comparison seems advisable to really assess the significance of adjustments in the IRMAE approach, as the VAE appears to be the more relevant competitor. - In addition, the effect of varying the number of extra linear matrices, l, is mostly not studied or discussed at all. Only in the first experiment on the synthetic dataset, two values for l are considered (l=2 and l=4). Does only a low number of additional linear matrices have a beneficial effect and increasing l has otherwise a detrimental effect on accuracy, generating samples etc.? Some more information here might be useful to assess the workings of the model, too.

Correctness: - All relevant details on the implementation, i.e. architecture and hyperparameters, are provided (in the supplement), allowing to reproduce the findings. The method and implementation is sound and empirical evaluation is provided to support the claims. - However, section "3.3 Downstream classification" might need so more details on what is displayed in table 2 and what exactly is tried to be shown with subsets of MNIST (see also the comment above in "weaknesses"). To me, this section is a bit unclear, but I might be missing something here.

Clarity: - In general, the submission is well-written and has a clear structure. However, I would consider moving “4 Related Work“ right before “2 Implicit Rank-Minimizing Autoencoder”. - Lines 65 – 68: As a suggestion to be more self-contained, one could elaborated more on why adding linear layers, i.e. linear mappings W1, W2, …, has an effect on rank minimisation. Being familiar with the work by Arora et al. and others (as cited in the submission), the claim is less surprising and also the ablation study experiments (section 3.4) give empirical evidence of that. But without prior knowledge on that matter this might appear counter-intuitive. In particular, as concatenating linear layers (l>1) is expected to be equivalent to one linear layer (l=1). - Table 2: The text and the caption speak of accuracy, however the table most likely shows the relative error, which should be stated more clearly.

Relation to Prior Work: - In my opinion related work is addressed appropriately.

Reproducibility: Yes

Additional Feedback: The submission presents an interesting idea and demonstrates convincingly desirable properties for a latent variable model. However, In the current form, I believe that the paper requires some more empirical evaluation and discussion with respect to VAE performance (as outlined above). Therefore, I am more inclined to see this submission rejected. # Minor Remarks: - L. 19: “leaerned” -> “learned” - Figure 4 caption: The legends display “singular values” as the labels, while the caption speaks of “eigenvalue”. - Figure 5 caption, last sentence: “IRAME” -> “IRMAE” ###################################################### # After Authors' Response: I appreciate the authors’ response. After careful reconsideration, I slightly changed my mind about the paper. I tend to agree with the assessment of Rev. 1, especially that the paper proposes a simple but potentially powerful idea which might be of a wide interest in the community. My first initial rating was "borderline" mainly due to some aspects and statements w.r.t. VAEs (e.g. ln. 107-109: "[...] comparable performances to the VAE while using a much lower effective-dimension code space"), that VAE comparison results were less compelling, and that the number of additional linear layers was not really discussed. However, in my opinion, the authors addressed these aspects satisfyingly in the rebuttal. In particular, I share the view of Rev. 1 that IRMAE performing comparably to VAE is an interesting result. The mechanism of identifying the relevant number of (intrinsic) latent dimensions is certainly interesting and might be relevant in different models, too. Therefore, I increase my rating to “accept” and advocate that the authors include the suggestions made by the different reviewers in the main text and include the additional experiments and figures in the supplement. With these changes to the initial version, I consider the paper relevant to be published at NeurIPS.

[Author Response · NeurIPS 2020]

We thank all reviewers for acknowledging the novelty and contributions of our work. We thank all reviewers for their
constructive comments for improving our paper. In this rebuttal, a) We improved our generative tasks experiments
including comparing IRMAE to modern AEs and comparing them with varying latent dimensions. b) We want to
emphasize the importance of the superior performance of our model on semi-supervised classification tasks. This shows
an advantage of our approach on representation learning for downstream tasks which was considered difficult for AEs.
As this is the first work of applying implicit regularization method, there could be many follow-up questions to explore.

**R1**: We added an experiment of our model using different initial variance settings. See Table 1 below. It's interesting
that the regularization effect varies corresponding to the initial condition. We will study this effect in our future work.

The ablation study in the appendix is to test whether tying linear matrices can help reduce the number of parameters,
which however results in worse performance. This shows the importance of having redundant degree of freedom for the
implicit regularization dynamics to function.

**R2**: Thanks for the experiment suggestions. We first added a comparison of our model to several modern AEs on
CelebA. See Table 2 below. Our model outperforms strong baselines such as WAE [1] and RAE [2]. We agree that
AEs perform differently with varying latent dimension. We compare IRMAE with AE with different latent dimension
settings in Table 4 and Figure 1 below. IRMAE outperforms AEs with optimal dimensionality.

We will reorganize the content as suggested. We will discuss deep linear generators papers in the related works.

**R3**: We added a new experiment of comparing our method against bottleneck AE in Table 4 and Figure 1 below. This
justifies our method over explicit low dimensional setting. We want to emphasize that using an explicitly selected latent
dimension requires prior knowledge. Our method, like many other regularization methods, does not guarantee finding
optimal latent dimension but reduces the effort of manually searching or requirement of prior knowledge.

The purpose of this work is to propose a genetic representation learning method instead of specific state-of-the-art
feature e.g. disentanglement by beta-VAE. Applying our method over these models will remain our future work.

Regarding L.143, ablation study: We fix the weight during training. This proves that the regularization effect comes
from the gradient descent dynamics instead of just the architecture.

We claim our method can have a stronger regularization effect by adding more linear layers. It does not guarantee
theoretical minimum rank. The number of linear layers is a hyperparameter that needs to be optimized. We admit we
lack enough experiments comparing the effect of different depths. Therefore, we added the experiment in Table 3 below.

The PCA experiment proves that IRMAE learns a dense latent space and solves the problem that naive deterministic
AEs have holes in their latent space.

**R4**: Regarding L.76-77, L.107-109, we agree that it's inappropriate to claim a superior performance related to smaller
intrinsic latent dimensions. VAE tends to use the entire prior latent space, while IRMAE, on the other hand, tends
to use smaller latent dimensions due to the regularization effect. It is possible that VAE with a proper selected latent
dimension can achieve better results. IRMAE and VAE have quite different mechanisms. And this is an quite interesting
phenomena of our approach compared to existing literature. Nonetheless, we believe a simple idea of inserting new
layers to achieve comparable results as widely-used VAE is a sufficient contribution.

Regarding L.131-132, IRMAE significantly outperforms VAE on low-data semi-supervised settings. These types of
tasks are important as AEs are usually considered less competitive in representation learning for downstream tasks [3].

We admit that we lack the comparison of different number of linear layers. Hence, we added a experiment in Table 3.

Table 1: Effect of different initial variance of linear matrices. MNIST.

| Variance | 1x | 2x | 4x |
|---|---|---|---|
| Latent Rank | 8 | 43 | 66 |
| FID | 37.4 | 33.8 | 49.0 |

Table 2: Effect of different number of linear layers. MNIST.

| Depth (l) | 2 | 4 | 8 | 12 |
|---|---|---|---|---|
| Latent Rank | 70 | 39 | 8 | 4 |
| FID | 44.0 | 30.1 | 37.4 | 62.6 |

Figure 1: IRMAE vs AE with varying latent dimension

Table 3: IRMAE vs modern AEs. FID on CelebA.

| WAE [1] | 53.7 |
|---|---|
| RAE [2] | 44.7 |
| IRMAE | **42.0** |

Table 4: IRMAE vs AE with different latent dimension. FID on CelebA.

| Latent dimension | 32 | 64 | 128 | 256 | 512 |
|---|---|---|---|---|---|
| IRMAE (l=4) | 81.6 | 64.6 | 47.6 | 42.7 | 42.0 |
| AE | 78.2 | 60.1 | 46.0 | 45.4 | 53.9 |

[1] "Wasserstein Auto-Encoders", I. Tolstikhin et al. ICLR 2018
[2] "From Variational To Deterministic Autoencoders" P. Ghosh et al. ICLR 2020
[3] "Large Scale Adversarial Representation Learning" J. Donahue et al. NeurIPS 2019


[Meta-Review · NeurIPS 2020]

This work got mixed reviews: R1 praised the potential impact of such a simple idea being shown to work remarkably well, but other reviewers had significant concerns about the empirical evaluation, which is especially important when the main contribution of the paper is to show that an idea is effective in practice. The reviewers were ultimately unable to reach a consensus about this paper, but all reviewers agreed that the core idea is promising, and R2, R3 and R4 raised their scores in light of the discussion and the author feedback. While the resulting scores still make this a difficult decision overall, I have chosen to recommend acceptance. The main point of discussion was whether the required changes to the manuscript require another review cycle or not. Indeed, the requested changes were quite broad: - demonstrate the effect of the initial variance of the linear layers - compare the model against modern autoencoder variants - compare against vanilla autoencoders with varying latent dimension - demonstrate the effect of the number of linear layers - avoid overclaiming, e.g. about the proposed model working well "with all types of optimizers" - etc. However, I think the authors have done a great job addressing the majority of these concerns in their rebuttal, which includes many new results. Given the potential impact of the idea and the prospect of follow-up work in various directions, I think accepting this work as it stands is worth the risk. In this decision, I am of course counting on the goodwill of the authors to prominently include the additional results from the rebuttal in the camera ready version of their manuscript, and to address any remaining concerns. Please make sure to also incorporate the reviewers' detailed feedback regarding typos and clarity.